# Bayesian inference of relative fitness on high-throughput pooled competition assays

**Manuel Razo-Mejia** [1]*, **Madhav Mani**[2,3], **Dmitri Petrov**[1,4,5]

**1** Department of Biology, Stanford University, Stanford, California, United States of America, **2** NSF-Simons Center for Quantitative Biology, Northwestern University, Chicago, Illinois, United States of America, **3** Department of Engineering Sciences and Applied Mathematics, Northwestern University, Chicago, Illinois, United States of America, **4** Stanford Cancer Institute, Stanford University School of Medicine, Stanford, California, United States of America, **5** Chan Zuckerberg Biohub, San Francisco, California, United States of America

\* mrazo@stanford.edu

## Abstract

The tracking of lineage frequencies via DNA barcode sequencing enables the quantification of microbial fitness. However, experimental noise coming from biotic and abiotic sources complicates the computation of a reliable inference. We present a Bayesian pipeline to infer relative microbial fitness from high-throughput lineage tracking assays. Our model accounts for multiple sources of noise and propagates uncertainties throughout all parameters in a systematic way. Furthermore, using modern variational inference methods based on automatic differentiation, we are able to scale the inference to a large number of unique barcodes. We extend this core model to analyze multi-environment assays, replicate experiments, and barcodes linked to genotypes. On simulations, our method recovers known parameters within posterior credible intervals. This work provides a generalizable Bayesian framework to analyze lineage tracking experiments. The accompanying open-source software library enables the adoption of principled statistical methods in experimental evolution.

## Author summary

In this study, we present a novel Bayesian pipeline for analyzing DNA barcode tracking sequencing data, addressing the challenge of accurately quantifying competitive microbial fitness in the presence of experimental noise. Our method uniquely contributes to understanding microbial evolutionary dynamics by enabling reliable inference of the relative fitness of diverse microbial strains from high-throughput lineage tracking assays. Our approach is distinct in its ability to systematically account for and propagate uncertainties from various noise sources throughout all inferred parameters. Furthermore, the error-propagation quality of our Bayesian method allows us to extend the inference pipeline to common dataset structures, such as jointly analyzing multiple experimental replicates or accounting for multiple unique barcodes mapping the equivalent genotypes. This comprehensive treatment of uncertainties is crucial in experimental settings where noise can significantly influence the results. Furthermore, we have optimized our pipeline for

**Data Availability Statement:** This paper is accompanied by a highly documented Julia software library—BarBay.jl (see documentation in https://mrazomej.github.io/BarBay.jl). Furthermore, to ensure transparency with every piece of

information presented in this paper, we have made all of the code used in the processing, analysis, and figure generation for this work also publicly available on this paper's GitHub repository (https://github.com/mrazomej/bayesian_fitness).

**Funding:** This work was supported by - The NIH/NIGMS, Genomics of rapid adaptation in the lab and in the wild R35GM11816506 (MIRA grant, to DP) - The NIH, Unravelling mechanisms of tumor suppression in lung cancer (R01CA23434903, to DP) - The NIH (PQ4), Quantitative and multiplexed analysis of gene function in cancer in vivo (R01CA23125303, to DP) - The NIH, Genetic Determinants of Tumor Growth and Drug Sensitivity in EGFR Mutant Lung Cancer (R01CA263715, to DP) - The NIH, Dissecting the interplay between aging, genotype, and the microenvironment in lung cancer (U01AG077922, to DP) - The NIH, Genetic dissection of oncogenic Kras signaling (R01CA230025, to DP) - The National Science Foundation-Simons Center for Quantitative Biology at Northwestern University and the Simons Foundation grant (597491, to MM) - The Chan Zuckerberg Initiative, an advised fund of Silicon Valley Community Foundation (DAF2023-329587, to MM) - MRM was supported by the Schmidt Science Fellowship. The funders had no role in study design, data collection and analysis, decision to publish, or preparation of the manuscript.

**Competing interests:** I have read the journal's policy and the authors of this manuscript have the following competing interests: MM is a Simons Investigator. DP is a CZ Biohub Investigator.

scalability, allowing it to handle large numbers of unique barcodes effectively. This scalability is essential for analyzing complex datasets typical in microbial fitness studies.

## Introduction

The advent of DNA barcoding—the ability to uniquely identify cell lineages with DNA sequences integrated at a specific locus—and high-throughput sequencing has opened new venues for understanding microbial evolutionary dynamics with an unprecedented level of temporal resolution [1–3]. These experimental efforts rely on our ability to reliably infer the relative fitness of an ensemble of diverse genotypes. Moreover, inferring these fitness values over an ensemble of environmental conditions can help us determine the phenotypic diversity of a rapid adaptation process [4].

As with any other sequencing-based quantification, tracking lineages via DNA barcode sequencing is inexorably accompanied by noise sources coming from experimental manipulation of the microbial cultures, DNA extraction, and sequencing library preparation that involves multiple rounds of PCR amplification, and the sequencing process itself. Thus, accounting for the uncertainty when inferring the relevant parameters from the data is a crucial step to draw reliable conclusions. Bayesian statistics presents a paradigm by which one can account for all known sources of uncertainty in a principled way [5]. This, combined with the development of modern Markov Chain Monte Carlo sampling algorithms [6] and approximate variational approaches [7] have boosted a resurgence of Bayesian methods in different fields [8].

We present a Bayesian inference pipeline to quantify the uncertainty about the parametric information we can extract from high-throughput competitive fitness assays given a model of the data generation process and experimental data. In these assays, the fitness of an ensemble of genotypes is determined relative to a reference genotype [3, 4]. Fig 1(A) shows a schematic of the experimental procedure in which an initial pool of barcoded strains are mixed with a reference strain and inoculated into fresh media. After some time—usually, enough time for the culture to saturate—an aliquot is transferred to fresh media, while the remaining culture is used for DNA sequencing of the lineage barcodes. The time-series information of the relative abundance of each lineage, i.e., the barcode frequency depicted in Fig 1(B), is used to infer the relative fitness—the growth advantage on a per-cycle basis—for each lineage with respect to the reference strain. The proposed statistical model accounts for multiple sources of uncertainty when inferring the lineages' relative fitness values (see Section "Experimental setup" for details on sources of uncertainty accounted for by the model). Furthermore, minor changes to the core statistical model allow us to account for relevant experimental variations of these competition assays. More specifically, in Section "Fitness inference on multiple environments", we present a variation of the statistical model to infer fitness on growth dilution cycles in multiple environments with proper error propagation. Furthermore, as described in Section "Accounting for experimental replicates via hierarchical models", our statistical model can account for batch-to-batch differences when jointly analyzing multiple experimental replicates using a Bayesian hierarchical model. Finally, a variant of these hierarchical models, presented in Section "*Accounting for multiple barcodes per genotype via hierarchical models*", can account for variability within multiple barcodes mapping to equivalent genotypes within the same experiment.

For all the model variations presented in this paper, we benchmark the ability of our pipeline to infer relative fitness parameters against synthetic data generated from logistic growth simulations with added random noise. A `Julia` package accompanies the present method to readily implement the inference pipeline with state-of-the-art scientific computing software.

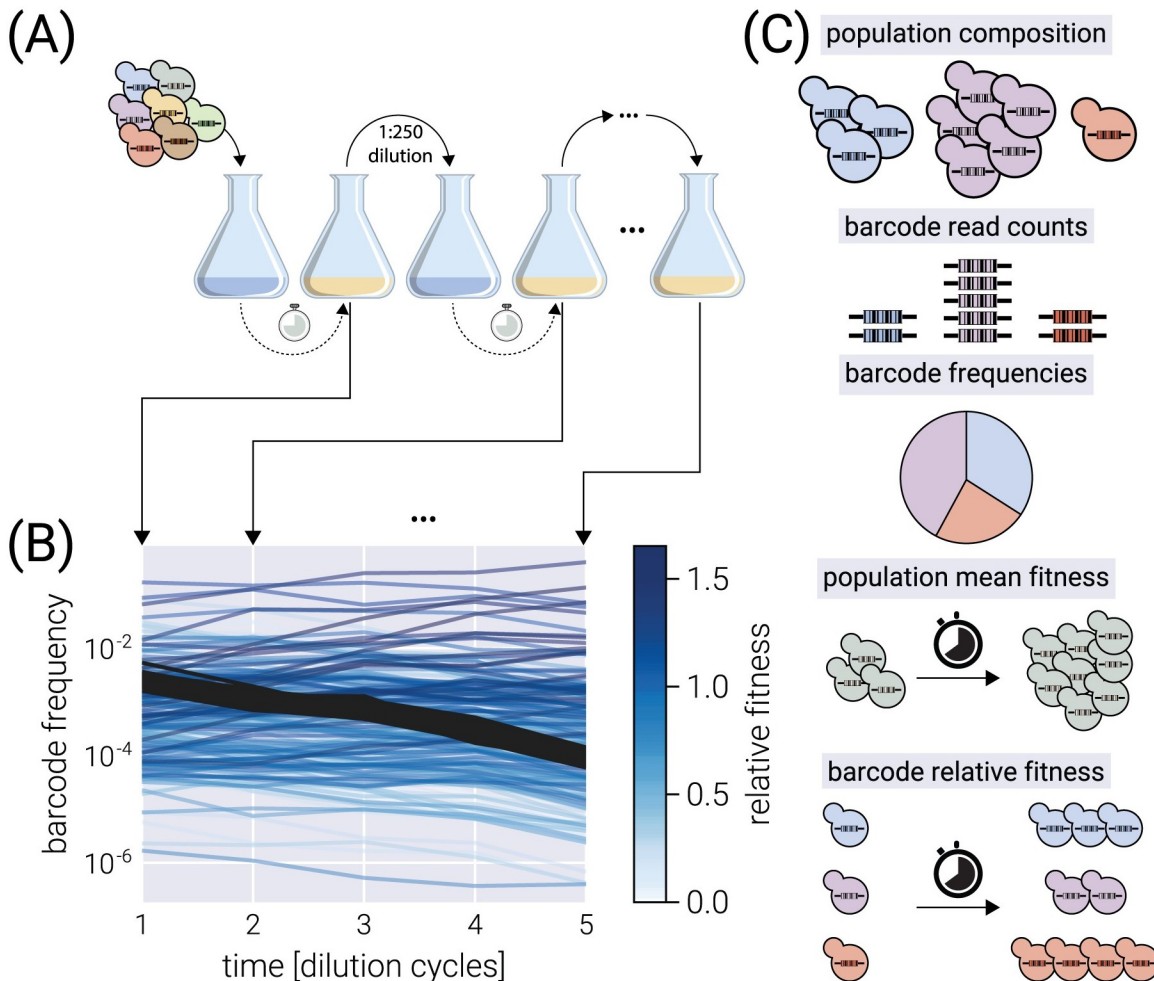

**Fig 1. Typical competitive fitness experiment.** (A) Schematic of the typical experimental design to determine the competitive fitness of an ensemble of barcoded genotypes. Genotypes are pooled together and grown over multiple growth-dilution cycles. At the end of each cycle, a sample is processed to generate a library for amplicon sequencing. (B) Typical barcode trajectory dataset. From each time point, the relative frequency of each barcode is determined from the total number of reads. Shades of blue represent different relative fitness. Darker gray lines define the typical trajectory of neutral lineages. (C) Sources of uncertainty accounted for by our method. The Bayesian model fit to the data propagates uncertainties from the categories schematically depicted to all parameters in the inference.

## Results

### Experimental setup

The present work is designed to analyze time-series data of relative abundance of multiple microbial lineages uniquely identified by a DNA barcode [3, 4]. In these competition assays, an ensemble of genotypes is pooled together with an unlabeled reference strain that, initially, represents the vast majority ($\geq$90%) of the cells in the culture (see schematic in Fig 1(A)). Furthermore, a fraction of labeled genotypes equivalent to the unlabeled reference strain—hereafter defined as *neutral* lineages—are spiked in at a relatively high abundance ($\approx 3 - 5\%$). The rest of the culture is left for the ensemble of genotypes of interest. This experimental design in which the barcodes of interest represent a small fraction of the culture serves three main purposes: First, the model used to infer the relative fitness differences between genotypes is valid in the regime where the genotype frequency is significantly smaller than one (see Section

"*Fitness model*"). Second, potential problems with frequency-dependent selection are minimized as long as the frequency of each genotype remains small. Third, having a single genotype dominating the culture standardizes the environment experienced by all genotypes. This is because variations in the chemistry of the environment are effectively dictated by the dominating genotype, allowing for reproducible growth-dilution cycles.

To determine the relative fitness of the ensemble of genotypes, a series of growth-dilution cycles are performed on either a single or multiple environments. In other words, the cultures are grown for some time; then, an aliquot is inoculated into fresh media for the next growth cycle. This process is repeated for roughly 4–7 cycles, depending on the initial abundances of the mutants and their relative growth rates. The DNA barcodes are sequenced at the end of each growth cycle to quantify the relative abundance of each of the barcodes. We point the reader to [4] for specific details on these assays for *S. cerevisiae* and to [3] for equivalent assays for *E. coli*. Fig 1(B) presents a typical barcode trajectory where the black trajectories represent the so-called *neutral lineages*, genetically equivalent to the untagged ancestor strain that initially dominates the culture. These spiked-in neutral lineages simplify the inference problem since the fitness metric of all relevant barcodes is quantified with respect to these barcodes—thus referred to as *relative fitness*.

## Preliminaries on mathematical notation

Before jumping directly into the Bayesian inference pipeline, let us establish the mathematical notation used throughout this paper. We define (column) vectors as underlined lowercase symbols such as

$$\underline{x} = \begin{bmatrix} x_1 \\ x_2 \\ \vdots \\ x_N \end{bmatrix}. \tag{1}$$

In the same way, we define matrices as double-underline uppercase symbols such as

$$\underline{\underline{A}} = \begin{bmatrix} A_{11} & A_{12} & \cdots & A_{1N} \\ A_{21} & A_{22} & \cdots & A_{2N} \\ \vdots & \vdots & \ddots & \vdots \\ A_{M1} & A_{M2} & \cdots & A_{MN} \end{bmatrix}. \tag{2}$$

## Fitness model

Empirically, each barcode frequency trajectory follows an exponential function of the form [1, 3, 4]

$$f_{t+1}^{(b)} = f_t^{(b)} e^{(s^{(b)} - \bar{s}_t)\tau}, \tag{3}$$

where $f_t^{(b)}$ is the frequency of barcode $b$ at the end of cycle number $t$, $s^{(b)}$ is the relative fitness with respect to the reference strain—the quantity we want to infer from the data—$\bar{s}_t$ is the

mean fitness of the culture at the end of cycle number $t$, and $\tau$ is the time pass between cycle $t$ and $t + 1$. We can rewrite Eq 3 as

$$\frac{1}{\tau}\ln\frac{f_{t+1}^{(b)}}{f_t^{(b)}} = (s^{(b)} - \bar{s}_t). \tag{4}$$

Eq 4 separates the measurements—the barcode frequencies—from the unobserved (sometimes referred to as latent) parameters we want to infer from the data—the population mean fitness and the barcode relative fitness. This is ultimately the functional form used in our inference pipeline. Therefore, the relative fitness is computed by knowing the log frequency ratio of each barcode throughout the growth-dilution cycles.

The presence of the neutral lineages facilitates the determination of the population mean fitness value $\bar{s}_t$. Since every relative fitness is determined relative to the neutral lineage that dominates the culture, we define their fitness to be $s^{(n)} = 0$, where the superscript $(n)$ specifies their neutrality. This means that Eq 4 for a neutral lineage takes the simpler form

$$\frac{1}{\tau}\ln\frac{f_{t+1}^{(n)}}{f_t^{(n)}} = -\bar{s}_t. \tag{5}$$

Therefore, we can use the data from these reference barcodes to directly infer the value of the population mean fitness.

It is important to notice that the frequencies $f_t^{(b)}$ are not the allele frequencies in the population (most of the culture is not sequenced since the reference strain is not barcoded), but rather the relative frequencies in the total number of sequencing reads. A way to conceptualize this subtle but important point is to assume exponential growth in the *number of cells $N_t^{(b)}$* of the form

$$N_{t+1}^{(b)} = N_t^{(b)}e^{\lambda^{(b)}\tau}, \tag{6}$$

for every barcode $b$ with growth rate $\lambda^{(b)}$. However, when we sequence barcodes, we do not directly measure the number of cells, but some number of reads $r_t^{(b)}$ that map to barcode $b$. In the simplest possible scenario, we assume

$$r_t^{(b)} \propto N_t^{(b)}, \tag{7}$$

where, importantly, the proportionality constant depends on the total number of reads for the library for cycle $t$, which might vary from library to library. Therefore, to compare the number of reads between libraries at different time points, we must normalize the number of reads to the same scale. The simplest form is to define a relative abundance, i.e., a frequency with respect to the total number of reads,

$$f_t^{(b)} \equiv \frac{r_t^{(b)}}{\sum_{b'} r_t^{(b')}}. \tag{8}$$

This is the frequency Eq 3 describes.

Our ultimate objective is to infer the relative fitness $s^{(b)}$ for each of the $M$ relevant barcodes in the experiment—hereafter referred to as $s^{(m)}$ to distinguish from the general $s^{(b)}$ and the neutral lineages $s^{(n)}$ relative fitness. To do so, we account for the three primary sources of uncertainty in our model:

1. Uncertainty in the determination of frequencies. Our model relates frequencies between adjacent growth-dilution cycles to the fitness of the corresponding strain. However, we do

not directly measure frequencies. Instead, our data for each barcode consists of a length $T$ vector of counts $\underline{r}^{(b)}$ for each of the $T$ cycles in which the measurements were taken.

2. Uncertainty in the value of the population mean fitness. We define neutral lineages to have fitness $s^{(n)} = 0$, helping us anchor the value of the population mean fitness $\bar{s}_t$ for each pair of adjacent growth cycles. Moreover, we take this parameter as an empirical parameter to be obtained from the data, meaning that we do not impose a functional form that relates $\bar{s}_t$ to $\bar{s}_{t+1}$. Thus, we must infer the $T-1$ values of this population mean fitness with their uncertainty that must be propagated to the value of the mutants' relative fitness.

3. Uncertainty in each of the mutants' fitness values.

Fig 1(C) shows schematically the sources of uncertainty accounted for by our model. The first three sources—population composition, barcode read counts, and barcode frequencies—all contribute to uncertainty source 1. Uncertainty source 2 and 3 are depicted by the last two sources, respectively. To account for all these sources of uncertainty in a principled way, in the next section, we develop a Bayesian inference pipeline.

## Bayesian inference

As defined in the previous section, our ultimate objective is to infer the vector of relative fitness values

$$\underline{s}^M = \left(s^{(1)}, s^{(2)}, \ldots, s^{(M)}\right)^{\dagger}, \tag{9}$$

where $^{\dagger}$ indicates the transpose. Our data consists of an $T \times B$ matrix $\underline{\underline{R}}$, where $B = M + N$ is the number of unique barcodes given by the sum of the number of unique, relevant barcodes we care about, $M$, and the number of unique neutral barcodes, $N$, and $T$ is the number of growth cycles where measurements were taken. The data matrix is then of the form

$$\underline{\underline{R}} = \begin{bmatrix} - & \underline{r}_1 & - \\ - & \underline{r}_2 & - \\ & \vdots & \\ - & \underline{r}_T & - \end{bmatrix}, \tag{10}$$

where each row $\underline{r}_t$ is a $B$-dimensional array containing the raw barcode counts at cycle $t$. We can further split each vector $\underline{r}_t$ into two vectors of the form

$$\underline{r}_t = \begin{bmatrix} \underline{r}_t^N \\ \underline{r}_t^M \end{bmatrix}, \tag{11}$$

i.e., the vector containing the neutral lineage barcode counts $\underline{r}_t^N$ and the corresponding vector containing the mutant barcode counts $\underline{r}_t^M$. Following the same logic, matrix $\underline{\underline{R}}$ can be split into two matrices as

$$\underline{\underline{R}} = [\underline{\underline{R}}^N \ \underline{\underline{R}}^M], \tag{12}$$

where $\underline{\underline{R}}^N$ is a $T \times N$ matrix with the barcode reads time series for each neutral lineage and $\underline{\underline{R}}^M$ is the equivalent $T \times M$ matrix for the non-neutral lineages.

Our objective is to compute the joint probability distribution for all relative fitness values given our data. We can express this joint posterior distribution using Bayes theorem as

$$\pi(\underline{s}^M \mid \underline{\underline{R}}) = \frac{\pi(\underline{\underline{R}} \mid \underline{s}^M)\pi(\underline{s}^M)}{\pi(\underline{\underline{R}})},\tag{13}$$

where hereafter $\pi(\cdot)$ defines a probability density function. When defining our statistical model, we need not to focus on the denominator on the right-hand side of Eq 13. Thus, we can write

$$\pi(\underline{s}^M \mid \underline{\underline{R}}) \propto \pi(\underline{\underline{R}} \mid \underline{s}^M)\pi(\underline{s}^M).\tag{14}$$

However, when implementing the model computationally, the normalization constant on the right-hand side of Eq 13 must be computed. This can be done from the definition of the model via an integral of the form

$$\pi(\underline{\underline{R}}) = \int d^M\underline{s}^M \pi(\underline{\underline{R}} \mid \underline{s}^M)\pi(\underline{s}^M),\tag{15}$$

also known as a marginalization integral. Hereafter, differentials of the form $d^n$ imply a $n$-dimensional integral.

Although Eqs 13 and 14 seem simple enough, recall that Eq 3 relates barcode frequency values and the population mean fitness to the mutant relative fitness. Therefore, we must include these nuisance parameters as part of our inference problem. We direct the reader to Section B of S1 Text for the exact definitions of these parameters. Here, it suffices to say that the inference problem must include the vector $\bar{\underline{s}}_T$ of all population mean fitness values and the matrix $\underline{\underline{F}}$ of all barcode frequencies within the sequencing data. With these nuisance variables in hand, the full inference problem we must solve takes the form

$$\pi(\underline{s}^M, \bar{\underline{s}}_T, \underline{\underline{F}} \mid \underline{\underline{R}}) \propto \pi(\underline{\underline{R}} \mid \underline{s}^M, \bar{\underline{s}}_T, \underline{\underline{F}})\pi(\underline{s}^M, \bar{\underline{s}}_T, \underline{\underline{F}}).\tag{16}$$

To recover the marginal distribution over the non-neutral barcodes relative fitness values, we can numerically integrate out all nuisance parameters, i.e.,

$$\pi(\underline{s}^M \mid \underline{\underline{R}}) = \int d^{T-1}\bar{\underline{s}}_T \int d^B\underline{f}_1 \cdots \int d^B\underline{f}_T \; \pi(\underline{s}^M, \bar{\underline{s}}_T, \underline{\underline{F}} \mid \underline{\underline{R}}).\tag{17}$$

**Factorizing the posterior distribution.** The left-hand side of Eq 16 is extremely difficult to work with. However, we can take advantage of the structure of our inference problem to rewrite it in a more manageable form. Specifically, the statistical dependencies of our observations and latent variables allow us to factorize the joint distribution into the product of multiple conditional distributions. To gain some intuition about this factorization, let us focus on the inference of the population mean fitness values $\bar{\underline{s}}_T$. Eq 5 relates the value of the population mean fitness to the neutral lineage frequencies and nothing else. This suggests that when writing the posterior for these population mean fitness parameters, we should be able to condition it only on the neutral lineage frequency values, i.e., $\pi(\bar{\underline{s}}_T \mid \underline{\underline{F}}^N)$. We point the reader to Section B in S1 Text for the full mathematical details on this factorization. For our purpose here, it suffices to say we can rewrite the joint probability distribution as a product of conditional

distributions of the form

$$\pi(\underline{s}^M, \bar{\underline{s}}_T, \underline{F} \mid \underline{R}) = \pi(\underline{s}^M \mid \bar{\underline{s}}_T, \underline{F}^M)\pi(\bar{\underline{s}}_T \mid \underline{F}^N)\pi(\underline{F} \mid \underline{R}). \qquad (18)$$

Written in this form, Eq 18 captures the three sources of uncertainty listed in Section "Fitness model" in each term. Starting from right to left, the first term on the right-hand side of Eq 18 accounts for the uncertainty when inferring the frequency values given the barcode reads. The second term accounts for the uncertainty in the values of the mean population fitness at different time points. The last term accounts for the uncertainty in the parameter we care about—the mutants' relative fitnesses. We refer the reader to Section B in S1 Text for an extended description of the model with specific functional forms for each term on the left-hand side of Eq 18 as well as the extension of the model to account for multiple experimental replicates or hierarchical genotypes.

**Variational inference.** One of the technical challenges to the adoption of Bayesian methods is the analytical intractability of integrals such as that of Eq 17. Furthermore, even though efficient Markov Chain Monte Carlo (MCMC) algorithms such as Hamiltonian Montecarlo can numerically perform this integration [6], the dimensionality of the problem in Eq 18 makes an MCMC-based approach prohibitively slow.

To overcome this computational limitation, we rely on the recent development of the automatic differentiation variational inference algorithm (ADVI) [7]. Briefly, when performing ADVI, our target posterior distribution $\pi(\theta \mid \underline{R})$, where $\theta = (\underline{s}^M, \bar{\underline{s}}_T, \underline{F})$, is replaced by an approximate posterior distribution $q_\phi(\theta)$, where $\phi$ fully parametrizes the approximate distribution. As further explained in Section A in S1 Text, the numerical integration problem is replaced by an optimization problem of the form

$$q_\phi^*(\theta) = \min_\phi D_{KL}(q_\phi(\theta)||\pi(\theta \mid \underline{R})), \qquad (19)$$

where $D_{KL}$ is the Kulback-Leibler divergence. In other words, the complicated high-dimensional numerical integration problem is transformed into a much simpler problem of finding the value of the parameters $\phi$ such that Eq 19 is satisfied as best as possible within some finite computation time. Although to compute Eq 19, we require the posterior distribution we are trying to approximate $\pi(\theta \mid \underline{R})$, it can be shown that maximizing the so-called evidence lower bound (ELBO) [9]—equivalent to minimizing the variational free energy [10]—is mathematically equivalent to performing the optimization prescribed by Eq 19. We direct the reader to Section A in S1 Text for a short primer on variational inference.

This work is accompanied by the Julia library `BarBay.jl` that makes use of the implementation of both MCMC-based integration as well as ADVI optimization to numerically approximate the solution of Eq 17 within the Julia ecosystem [11].

## Inference on a single dataset

To assess the inference pipeline performance, we applied it to a simulated dataset with known ground truth relative fitness values (See Section D in S1 Text for details on simulation). Fig 2 (A) shows the structure of the synthetic dataset. The majority of barcodes of interest (faint color lines) are adaptive compared to the neutral barcodes ($s^{(m)} > 0$). Although the barcode frequency trajectories look relatively smooth, our fitness model requires the computation of the log frequency ratio between adjacent time points as derived in Eq 4. Fig 2(B) shows such data transformation where we can better appreciate the observational noise input into our statistical model. This noise is evident for the darker lines representing the neutral barcodes since all of these lineages are assumed to be identically distributed.

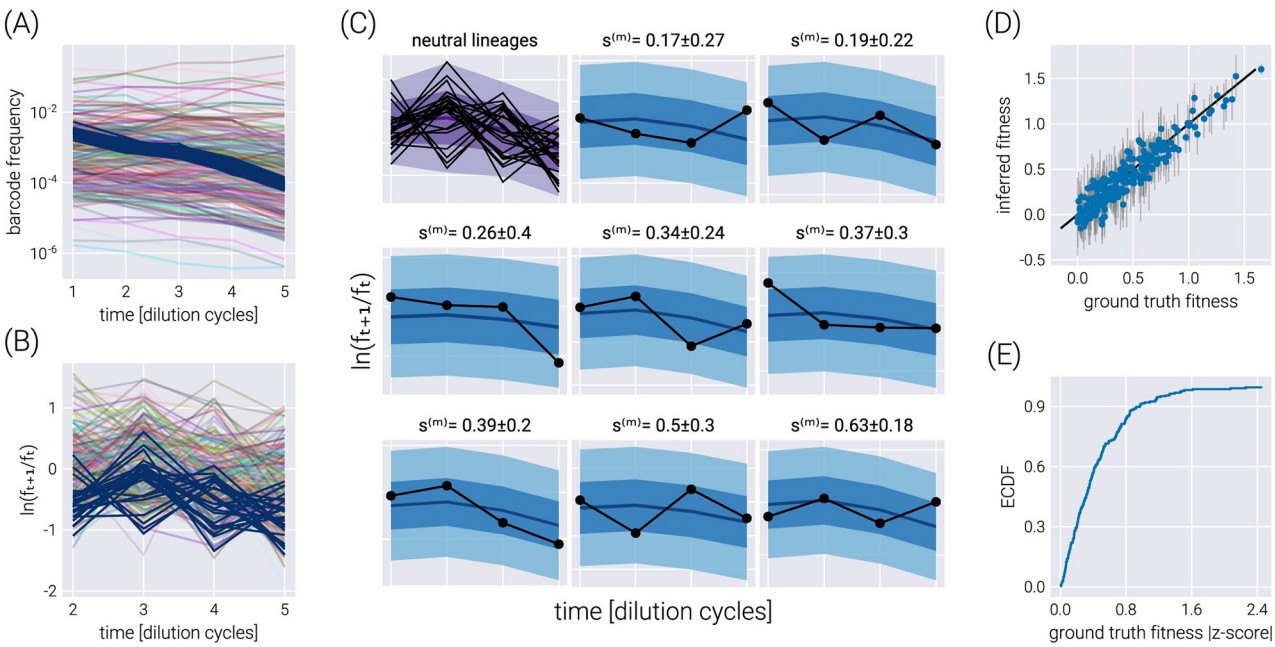

**Fig 2. Single dataset inference.** (A) Frequency trajectories that represent the raw data going into the inference. (B) Log frequency ratio between two adjacent time points used by the inference pipeline. Darker lines represent the neutral barcodes. These transformed data are much more noisy than the seemingly smooth frequency trajectories. (C) Examples of the posterior predictive checks for all neutral lineages (upper left panel) and a subset of representative mutant lineages. Shaded regions represent the 95%, 68%, and 5% credible regions for the data. The reported errors above the plot represent the 68% credible region on the mutant relative fitness marginal distribution. (D) Comparison between the ground truth fitness value from the logistic-growth simulation and the inferred fitness value. Gray error bars represent the 68% posterior credible region for the relative fitness values. (E) The empirical cumulative distribution function (ECDF) for the absolute z-score value of the ground truth parameter value within the inferred fitness posterior distribution.

To visualize the performance of our inference pipeline in fitting our fitness model to the observed data, we compute the so-called posterior predictive checks (PPC). In short, the PPC consists of repeatedly generating synthetic datasets in agreement with the results from the inference results. In other words, we use the resulting parameter values from the ADVI inference to generate possible datasets in agreement with the inferred values (See Section C in S1 Text for further details on these computations). Fig 2(C) shows these results for all neutral lineages (upper left corner plot) and a few representative non-neutral barcodes. The different color shades represent the 95%, 68%, and 5% credible regions, i.e., the regions where we expect to find the data with the corresponding probability—or in terms of our parameter, the *X*% credible region is the interval where we expect the true parameter value to lie with *X*% probability.

The main advantage of our method is the natural interpretability of these credible regions where an *X*% credible region indeed captures the region of parameter space where we expect with *X*% probability the actual value of the parameter lies given our statistical model, our prior information, and the observed experimental data. Bayesian methods avoid common misconceptions associated with the construction of frequentists confidence intervals [12].

To capture the global performance of the model, Fig 2(D) compares the known ground truth with the inferred relative fitness value for all barcodes of interest. There is an excellent degree of correspondence between these values, with the error bars representing the 68% credible region for the parameter value crossing the identity line for most barcodes. This latter

point is made clear with Fig 2(E) where ≈90% of ground truth fitness values fall within one standard deviation of the mean in the inferred posterior distributions.

## Fitness inference on multiple environments

The fitness model in Eq 3 relates nuisance parameters such as the population mean fitness and the barcode frequencies to the relative fitness parameter we want to infer from the data. These dependencies imply that uncertainty on the estimates of these nuisance parameters influences the inference of the relevant parameters. For example, imagine a scenario where the neutral lineages data were incredibly noisy, leading to poor estimates of the population mean fitness values $\bar{\underline{s}}_T$. Since the relative fitness of any non-neutral barcode $s^{(m)}$ is determined with respect to these neutral barcodes, not accounting for the lack of precision in the value of the population mean fitness would result in misleading estimates of the accuracy with which we determine the value of the parameter we care about. Thus, propagating these sources of uncertainty in nuisance parameters is vital to generate an unbiased estimate of the relevant information we want to extract from the data. One of the benefits of Bayesian methods is the intrinsic error propagation embedded in the mathematical framework. For our previous example, the uncertainty on the value of the population mean fitness values is propagated to the relative fitness of a non-neutral barcode since we defined a joint posterior distribution over all parameters as fully expressed in Eq 16.

This natural error propagation can help us with the experimental design schematized in Fig 3(A). Here, rather than performing growth-dilution cycles in the same environment, the cells are diluted into a different environment. Thus, the uncertainty on the fitness estimate for the previous environment must be propagated to that of the next one. To validate the extension of our statistical model to this scenario, Fig 3(B) shows the trajectory of the log frequency ratios between adjacent time points. The different colored regions correspond to the different environments. For this simulation, the growth rate of Environment 2 was set to be, on average, half of the average growth rate in Environment 1. Equivalently, the growth rate in Environment 3 was set to be, on average, twice the average growth rate in Environment 1. Fig 3(C)–3(E) show the correspondence between the simulation ground truth and the inferred fitness values, where the error bars represent the 68% credible region. Fig 3(F) summarizes the performance of our inference pipeline by showing the empirical cumulative distribution functions for the absolute value of the ground truth fitness value z-score within the posterior distribution. This plot shows that, overall, ≈75% of inferred mean values fall within one standard deviation of the ground truth. For completeness, Fig 3(G) shows the posterior predictive checks for a few example barcodes.

## Accounting for experimental replicates via hierarchical models

Our inference pipeline can be extended to account for multiple experimental replicates via Bayesian hierarchical models [13]. Briefly, when accounting for multiple repeated measurements of the same phenomena, there are two extreme cases one can use to perform the data analysis: On the one hand, we can treat each measurement as entirely independent, losing the power to utilize multiple measurements when trying to learn a single parameter. This can negatively impact the inference since, in principle, the value of our parameter of interest should not depend on the particular experimental replicate in question. However, this approach does not allow us to properly "combine" the uncertainties in both experiments when performing the inference. On the other hand, we can pool all data together and treat our different experiments as a single measurement with higher coverage. However, by doing so, we lose all information about experiment-to-experiment variability due to intrinsic biological variability and

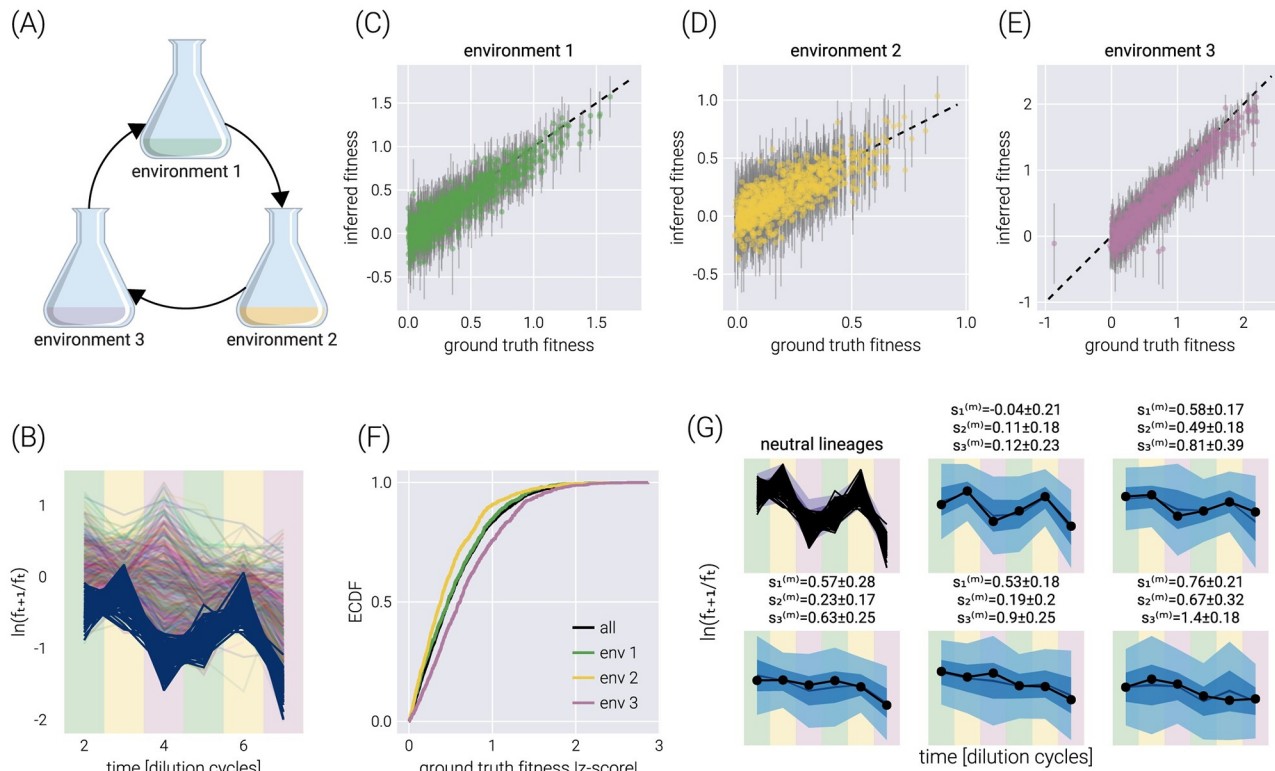

**Fig 3. Multi-environment fitness inference.** (A) Schematic of the simulated experimental design where growth-dilution cycles are performed into different environments for each cycle. (B) log frequency rations between adjacent time points. Darker lines represent the neutral barcodes. The colors in the background demark the corresponding environment, matching colors in (A). Environment 2 is set to have, on average, half the growth rate of environment 1. Likewise, environment 3 is set to have, on average, twice the growth rate of environment 1. (C-E) Comparison between the ground truth fitness value from the logistic-growth simulation and the inferred fitness value for each environment. Gray error bars represent the 68% posterior credible region. (F) The empirical cumulative distribution function (ECDF) for the absolute z-score value of the ground truth parameter value within the inferred fitness posterior distribution for all fitness values (black line) and each environment individually (color lines). (G) Examples of the posterior predictive checks for all neutral lineages (upper left panel) and a subset of representative mutant lineages. Shaded regions surrounding the data represent the 95%, 68%, and 5% credible regions for the data. The reported errors above the plot represent the 68% credible region on the mutant relative fitness marginal distribution. Background colors match those of (A).

environmental fluctuations—such as small temperature fluctuations—that we cannot control. In this sense, combining the datasets defeats the purpose of performing multiple measurements to account for this variability when extracting the relevant information from the observations.

Hierarchical models present a middle ground between these extremes. First, hierarchical models rely on the definition of so-called *hyper-parameters*, that capture the parametric inference we are interested in—for this inference problem, we have a hyper-fitness value $\theta^{(m)}$ for each non-neutral barcode. Second, each experiment draws randomly from the distribution of this hyper-parameter, allowing for subtle variability between experiments to be accounted for—in the present inference pipeline, each experimental replicate gets assigned a *local* fitness value $s_i^{(m)}$, where the extra sub-index indicates the *i*-th experimental replicate. Conceptually, we can think of the local fitness for replicate *i* as being sampled from a distribution that depends on the value of the global hyper-fitness value, i.e., $s_i^{(m)} \sim \pi_{\theta^{(m)}}$, where the subindex $\theta^{(m)}$ indicates the distribution's parametric dependence on the hyper-fitness value. This way of interpreting the connection between the distribution $\pi_{\theta^{(m)}}$ and the local fitness implies that a

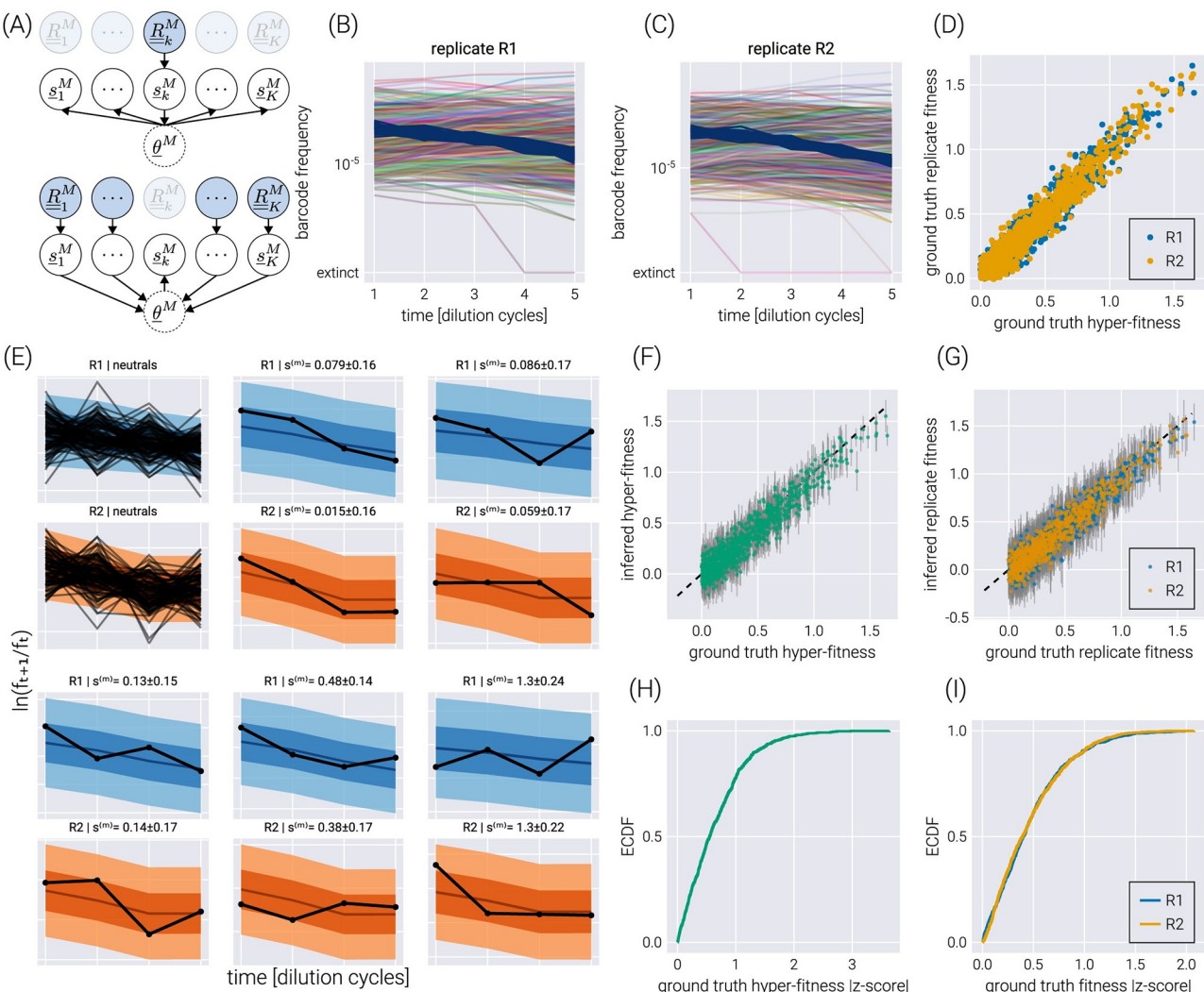

**Fig 4. Hierarchical model on experimental replicates.** (A) Schematic depiction of the interactions between local fitness values $\underline{s}_k^M$ through the global hyper-fitness value $\underline{\theta}^M$ for $K$ hypothetical experimental replicates. The upper diagram shows how the data from replicate $k$ informs all local fitness values via the hyper-fitness parameter. The lower panel shows the reverse, where all other datasets inform the local fitness value. (B-C) Simulated replicate datasets with 900 barcodes of interest and 100 neutral lineages. (D) Comparison between the simulation ground truth hyper-fitness and each replicate ground truth fitness. The scatter between parameters captures experimental batch effects. (E) Examples of the posterior predictive checks for all neutral lineages (upper left panels) and a subset of representative mutant lineages. Shaded regions from light to dark represent the 95%, 68%, and 5% credible regions. (F-G) Comparison between the simulation's ground truth hyper-fitness (F) and replicate fitness (G) values and the inferred parameters. Gray error bars represent the 68% posterior credible region. (H-I) The empirical cumulative distribution function (ECDF) for the absolute z-score value of the ground truth parameter value within the inferred hyper-fitness posterior distribution (H) and replicate fitness (I).

large replicate-to-replicate variability would lead to a broad hyper-fitness distribution—implying a large uncertainty when determining the parameter that characterizes the overall relative fitness. We point the reader to Section B.4 in S1 Text for the full definition of the hierarchical model used in this section. Importantly, as schematized in Fig 4(A), the influence between different experimental replicates runs both ways. First, the data from one experimental replicate ($\underline{\underline{R}}_k^M$ in the diagram) informs all local fitness values via the global hyper-fitness (upper panel in Fig 4(A)). Second, the local fitness value is informed by the data from all experimental replicates via the same global hyper-fitness parameter (lower panel in Fig 4(A)).

To test the performance of this model, we simulated two experimental replicates with 1000 unique barcodes (see Fig 4(B) and 4(C)) where we randomly sampled a ground truth hyper-fitness value $\theta^{(m)}$ for each barcode. We sampled a variation from this hyper-fitness value for each experimental replicate $s_i^{(m)}$ to capture experimental batch effects. Fig 4(D) shows the relationship between hyper-fitness and replicate fitness values for this simulation. The spread around the identity line represents the expected batch-to-batch variation. The posterior predictive checks examples in Fig 4(E) show that the hierarchical model can correctly fit the data for each experimental replicate. Furthermore, Fig 4(F) and 4(G) show a high correlation between the ground truth and the inferred fitness values. The empirical cumulative distribution functions shown in Fig 4(H) and 4(I) reveal that for ≈75% of the non-neutral barcodes, the ground truth hyper-fitness values fall within one standard deviation from the mean value in the posterior distributions.

As shown in Fig 5, the structure imposed by the hierarchical model schematized in Fig 4 (A), where we explicitly account for the connection between experimental replicates can improve the quality of the inference. Inferred fitness values between experimental replicates exhibit a stronger degree of correlation in the hierarchical model (Fig 5(A)) compared to conducting inference on each replicate independently (Fig 5(B)). Moreover, when comparing the inferred hyper-fitness values—the objective parameter when performing multiple experimental measurements—the hierarchical model outperforms averaging the independent experimental replicates as shown in Fig 5(C) and 5(D).

## Accounting for multiple barcodes per genotype via hierarchical models

Hierarchical models can also capture another experimental design in which multiple barcodes map to the same or an equivalent genotype. As we will show, this many-to-one mapping can improve the inference compared to the extreme cases of inferring the fitness of each barcode independently or pooling the data of all barcodes mapping to a single genotype. As schematized in Fig 6(A), a small modification of the base model allows us to map the structure of our original model to that of a hierarchical model with a fitness hyperparameter vector $\underline{\theta}^G$, where $G$ is the number of genotypes in the dataset. This analysis pipeline assumes a known barcode-to-genotype mapping to assign each barcode to the corresponding hyper-fitness parameter uniquely.

Fig 6(B) shows a single experimental replicate in which 90 genotypes were assigned a random number of barcodes (a multinomial distribution with a mean of ten barcodes per genotype) for a total of 900 non-neutral barcodes. To assess the performance of the hierarchical model proposed in Fig 6(A), we performed inference using this hierarchical model, as well as the two extreme cases of ignoring the connection between the barcodes belonging to the same genotype—equivalent to performing inference using the model presented in Fig 2(A) over the barcodes—or pooling the data of all barcodes belonging to the same genotype into a single count—equivalent to performing inference using the model presented in Fig 2(A) over the pooled barcodes. Fig 6(C) and 6(D) shows the comparison between the simulation ground truth and the inferred values for these three cases. Not only do the hierarchical model results show higher degrees of correlation with the ground truth, but the error bars (representing the 68% credible regions) are smaller, meaning that the uncertainty in the estimate of the parameter we care about decreases when using the hierarchical model. The improvement in the prediction can be seen in Fig 6(F) where the empirical cumulative distribution function of the absolute difference between the mean inferred value and the simulation ground truth is shown for all three inference models. The hierarchical model's curve ascends more rapidly, showing that, in general, the inferred values are closer to the ground truth. For completeness, Fig 6(G)

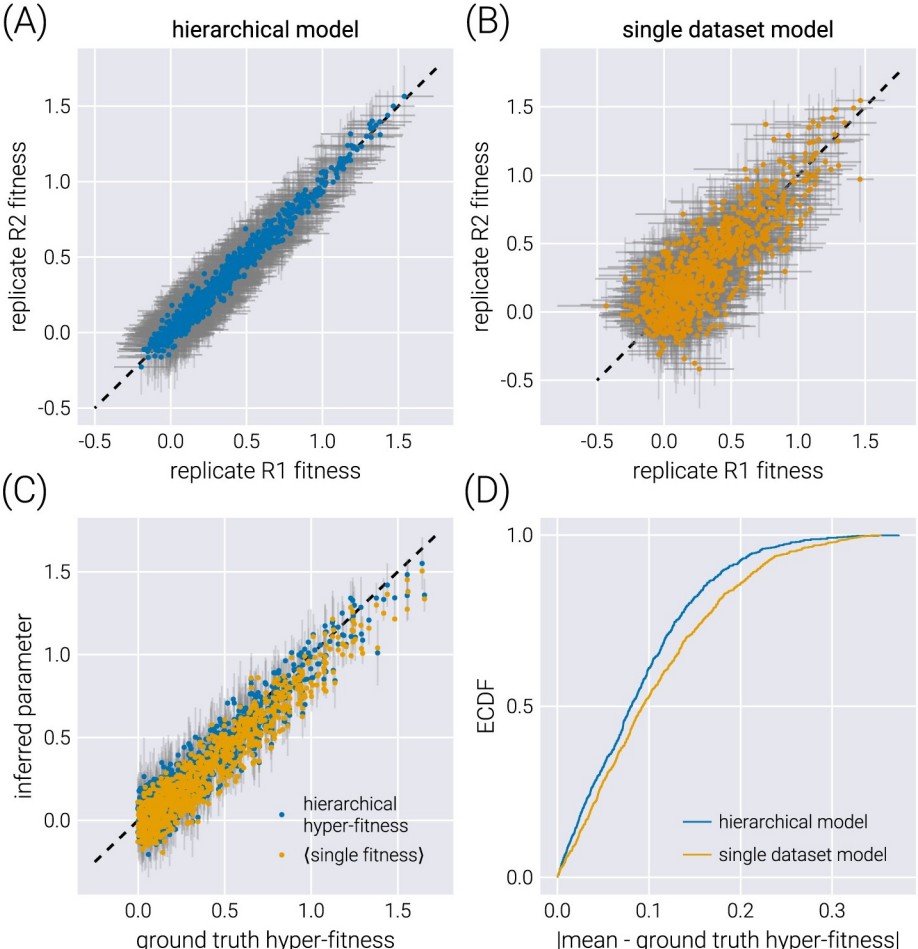

**Fig 5. Comparison between hierarchical model and single dataset model.** (A-B) comparison of inferred fitness values between experimental replicates when fitting a hierarchical model (A) or independently fitting each dataset (B). Gray error bars represent the 68% credible regions. (C) Comparison between the ground truth hyper-fitness value and the inferred parameters. The blue dots show the inferred hyper-fitness values when assuming a hierarchical model. Gray error bars show the 68% credible region for this inference. The yellow dots show the average of the mean inferred fitness values for the two experimental replicates. No error bars are shown for these, as it is inappropriate to compute one with two data points per non-neutral barcode. (D) Empirical cumulative distribution function (ECDF) of the absolute difference between the inferred mean and the ground truth hyper-fitness.

shows some examples of how the hierarchical model can capture the raw log-frequency count observations.

## Discussion

Experimental evolution of microbial systems has dramatically advanced our understanding of the basic principles of biological evolution [14]. From questions related to the optimal fine-tuning of gene expression programs [15], to the dimensionality, geometry, and accessibility of the adaptive fitness landscape explored by these rapidly adapting populations [4, 16], to the emergence of eco-evolutionary dynamics in a long-term evolution experiment [17]; for all of these and other cases, the microbial experimental platform combined with high-throughput sequencing has been essential to tackling these questions with empirical data. This exciting

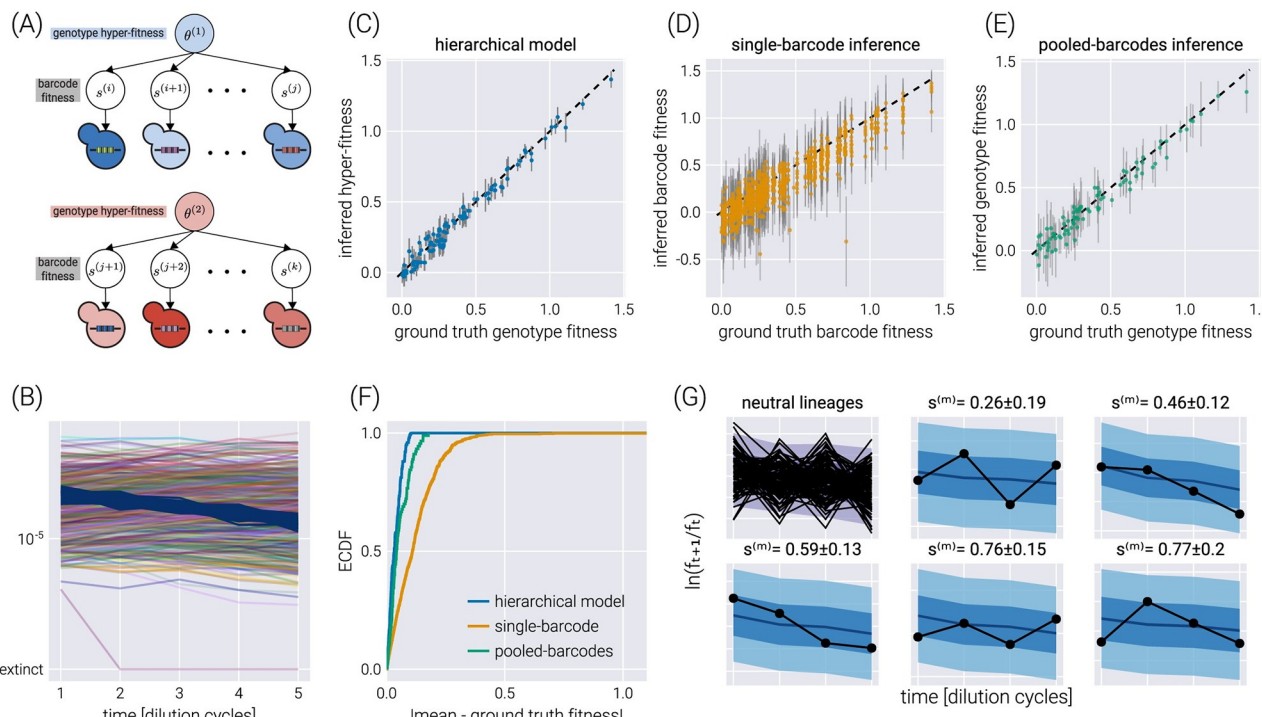

**Fig 6. Hierarchical model for multiple barcodes per genotype.** (A) Schematic depiction of the hierarchical structure for multiple barcodes mapping to a single genotype. A set of barcodes mapping to an equivalent genotype map to "local" fitness values $s^{(b)}$ that are connected via a hyper-fitness parameter for the genotype $\theta^{(g)}$. (B) Simulated dataset with 100 neutral lineages and 900 barcodes of interest distributed among 90 genotypes. (C-E) Comparison between the inferred and ground truth fitness values for a hierarchical model (C), a model where each barcode is inferred independently (D), and a model where barcodes mapping to the same genotype are pooled together (E). Gray error bars represent the 68% credible regions. (F) Empirical cumulative distribution function (ECDF) of the absolute difference between the inferred mean and the ground truth fitness values for all three models. (G) Examples of the posterior predictive checks for all neutral lineages (upper left panels) and a subset of representative mutant lineages. Shaded regions from light to dark represent the 95%, 68%, and 5% credible regions.

research area promises to improve as new culturing technologies [18] as well as more complex lineage barcoding schemes [2, 19], are adopted.

For this data-heavy field, properly accounting for the uncertainty in parameters inferred from experiments is vital to ensure the conclusions drawn are reliable. Bayesian statistics presents a principled way to quantify this uncertainty systematically [20]. Moreover, Bayesian analysis offers a more natural way to interpret the role that probability theory plays when performing data analysis compared to the often-misinterpreted frequentist methods [21]. Nevertheless, the technical challenges associated with Bayesian analysis has limited its application. This is set to change as recognition of the misuse of frequentist concepts such as the p-value is receiving more attention [22]. Moreover, advances in numerical methods such as Hamiltonian Monte Carlo [6] and variational inference [7] allows for complex Bayesian models to be fit to empirical data.

In this paper, we present a computational pipeline to analyze lineage-tracking time-series data for massive-parallel competition assays. More specifically, we fit a Bayesian model to infer the fitness of multiple genotypes relative to a reference [3, 4]. The proposed model accounts for multiple sources of uncertainty with proper error propagation intrinsic to Bayesian methods. To scale the inference pipeline to large datasets with >10, 000 barcodes, we use the ADVI algorithm [7] to fit a variational posterior distribution. The main difference between our method and previous inference pipelines, such as [23], is that the present analysis provides

interpretable errors on the inferred fitness values. The reported uncertainty intervals—known as credible regions—can be formally interpreted as capturing the corresponding probability mass of finding the true value of the parameter given the model, the prior information, and the data. Furthermore, minor modifications to the structure of the statistical model presented in this work allow for the analysis of different experimental designs, such as growth-dilution cycles in different environments, joint analysis of multiple experimental replicates of the same experiment via hierarchical models, and a hierarchical model for multiple barcodes mapping to equivalent genotypes. We validate our analysis pipeline on simulated datasets with known ground truth, showing that the model fits the data adequately, capturing the ground truth parameters within the posterior distribution.

It is important to highlight some of the consequences of the general experimental design and the implicit assumptions within the proposed statistical model to analyze the resulting data. First, the composition of the population is such that the initial fraction of the population occupied by the barcoded genotypes is small—usually >90% of the initial population is the non-labeled reference strain. This constraint is important as the fitness model used to fit the time series data assumes that the tracked frequencies are $\ll 1$. Second, when computing log frequency ratios, we can run into the issue of dividing by zero. This is a common problem when dealing with molecular count data [24]. Our model gets around this issue by assuming that the frequency of any barcode cannot be, but still can get arbitrarily close to, zero. Therefore, we implicitly assume that no lineage goes extinct during the experiment. Moreover, the statistical model directly accounts for the uncertainty associated with having zero barcode counts, increasing the corresponding uncertainty. Third, the models presented in this paper require the existence of a labeled sub-population of barcoded reference strains. These barcodes help determine the fitness baseline, as every fitness is quantified with respect to this reference genotype. This experimental design constraint facilitates the inference of the population mean fitness since most of the culture—the unlabeled reference genotype—is not tracked. Fourth, the fitness model in Eq 3 does not assume any functional form for the cellular growth dynamics. This description of the changes in the relative frequency of the genotypes can approximate the dynamics for multiple growth forms. For example, in Section E in S1 Text, we apply our inference pipeline to simulations like those presented in [23] with non-logistic growth models. Moreover, Section F in S1 Text, in the supplementary materials, reanalyzes experimental data from [4], where previous work showed that the fitness advantage from some of the mutants results from a shorter lag phase at the beginning of the growth cycle [25]. Thus, our method's inferred relative fitness is a coarse-grained quantity that integrates the differences in fitness across the entire growth cycle as long as frequencies and frequency changes are small enough to be approximated by Eq 3. Finally, the presented statistical model assumes that relative fitness is solely a constant of the environment and the genotype. Future directions of this work could extend the fitness model to properly analyze data with time-varying or frequency-dependent fitness values.

In total, the statistical model presented in this work and the software package accompanying the paper allow for a principled way of quantifying the accuracy with which we can extract relevant parametric information from large-scale multiplexed fitness competition assays. Furthermore, the implementation of Bayesian models and their fitting via automatic differentiation approaches opens the gate to extend this type of formal analysis to the data-rich literature in experimental evolution and other high-throughput technologies applications.

## Materials and methods

All synthetic data generation and custom scripts used in this work were stored using Git version control. Code for analysis and figure generation can be found on the GitHub repository

([https://github.com/mrazomej/bayesian_fitness](https://github.com/mrazomej/bayesian_fitness)). The accompanying software package `BarBay.jl` can be directly installed from the `Julia` package repository, or via cloning the corresponding GitHub repository ([https://github.com/mrazomej/BarBay.jl](https://github.com/mrazomej/BarBay.jl)).

## Supporting information

**S1 Text. Supplementary materials.** Section A gives a short primer on variational inference. Section B defines the probabilistic models used throughout the main text. Section C details how the validity of the model is computed via posterior predictive checks. Section D explains how the simulated frequency trajectories are generated. Section E compares the inferences of our method with state-of-the-art methods in the literature. Section F reanalyzes experimental data from yeast evolution experiments. Section G details how the computation time scales with the number of barcodes.
(PDF)

## Acknowledgments

We would like to thank Griffin Chure and Michael Betancourt for their helpful advice and discussion. We would like to thank Karna Gowda, Spencer Farrell, and Shaili Mathur for critical observations on the manuscript. We are especially thankful to Grant Kinsler for kindly providing raw experimental data as well as lengthy discussions about the state-of-the-art inference method.

## Author Contributions

**Conceptualization:** Manuel Razo-Mejia.

**Data curation:** Manuel Razo-Mejia.

**Formal analysis:** Manuel Razo-Mejia.

**Funding acquisition:** Madhav Mani, Dmitri Petrov.

**Investigation:** Manuel Razo-Mejia.

**Methodology:** Manuel Razo-Mejia.

**Project administration:** Manuel Razo-Mejia, Madhav Mani.

**Software:** Manuel Razo-Mejia.

**Supervision:** Manuel Razo-Mejia, Madhav Mani, Dmitri Petrov.

**Validation:** Manuel Razo-Mejia.

**Visualization:** Manuel Razo-Mejia.

**Writing – original draft:** Manuel Razo-Mejia, Madhav Mani, Dmitri Petrov.

**Writing – review & editing:** Manuel Razo-Mejia, Madhav Mani.

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
