## [Decision Letter · Decision Letter 0]

15 Nov 2023

Dear Dr. Razo-Mejia,

Thank you very much for submitting your manuscript "Bayesian inference of relative fitness on high-throughput pooled competition assays" for consideration at PLOS Computational Biology.

As with all papers reviewed by the journal, your manuscript was reviewed by members of the editorial board and by several independent reviewers. In light of the reviews (below this email), we would like to invite the resubmission of a significantly-revised version that takes into account the reviewers' comments.

Two major issues raised by several reviewers are that the algorithm has not been tested on experimental data and has not been compared to currently available algorithms. Addressing at least one of these issues is required for publication.

We cannot make any decision about publication until we have seen the revised manuscript and your response to the reviewers' comments. Your revised manuscript is also likely to be sent to reviewers for further evaluation.

Sincerely,

Sergei Maslov

Academic Editor

PLOS Computational Biology

Zhaolei Zhang

Section Editor

PLOS Computational Biology

Reviewer's Responses to Questions

**Comments to the Authors:**

Reviewer #1: Review is uploaded as an attachment

Reviewer #2: By tracking lineages frequency via DNA barcodes in competitive cultures, it is possible to measure microbial fitness and phenotypic diversity on a large scale. However, many biological and non-biological noises will have a significant impact on the relationship between barcodes and phenotypes (fitness). Accordingly, the authors of this manuscript developed a Bayesian model-based pipeline that takes into account some of these uncertainties in the experimental setup described above. Additionally, this model was applied to analyze simulated multi-environment and replicate (by batches, or barcodes of the same genotype) experiments. As a whole, the manuscript has a solid theoretical basis and should be of general interest to the field. I believe that additional work is necessary before it can be published. My concerns are listed below.

== MAJOR ==

1. The manuscript contains a little bit too many technical details, so that readers might become distracted from the main logic flow. As an example, I do not believe it is necessary to comment on frequentist confidence intervals versus Bayesian credible regions starting at line 247. However, I lack a strong statistical background to provide a more subjective or comprehensive assessment or to say that all the technical details are trivial/unnecessary. Other reviewers with expertise in that area may be able to provide better suggestions. Nonetheless, moving some details to supplementary to better emphasize the main logic seems reasonable.

2. There is no description of the detailed simulation procedure in the main text or in SI. This would make reproducing the results difficult. The reader must also understand the factors that have been taken into consideration during the simulation in order to determine (i) whether the results presented are related to the reader's own experimental environment or not, and (ii) whether the performance assessment based on the simulation is just expected since the simulation entails exactly the same type of noise as those considered in the inference model. I understand from the manuscript that (ii) is exactly the case. I am not saying this approach is wrong, but it certainly needs clarification. Even better, sometimes the simulation may include some additional noise or perturbation that the inference model is unable to account for, so that one can determine whether the inference is robust against those specific additional noises or perturbations.

3. There is a disconnect between the results and the actual experimental data. In all tests, inferences are drawn based on simulated data. I understand that it gives the authors the ground truth fitness. Nevertheless, every simulation is based on some simplification or assumption that may not be valid in reality. It is difficult to determine the relevance of the method without actually relating the model to real data. I would like to provide two more specific comments in this regard.

3.1. The authors seem to assume some specific composition of the competing population (e.g. most strains are not barcoded), as well as some specific distribution of mutational fitness effects (s) (most barcodes are slightly more adaptive than wild-type). In my opinion, both assumptions are frequently violated. For example, in https://www.science.org/doi/10.1126/science.aae0568, https://www.nature.com/articles/nature17995 and https://academic.oup.com/mbe/article/39/5/msac086/6575838, all genotypes are barcoded. In the commonly used "deep mutational scanning" assays, where only proximal mutations are tested, most mutants (containing only a few simple mutations) have fitness very similar to that of the wild-type. Can the model accurately estimate their fitness? What is the significance of their differences with wild-type?

3.2. There should be some analysis based on actual experimental data (such as those in the three papers mentioned above). A straightforward test would be to determine whether the Bayesian-inferred fitness shows better between-replicate-correlations than the more naively estimated fitness reported in those papers. I am certain that the author can come up with more analyses relating the actual experimental data to the accuracy of their methodology.

==Minor==

1. The schematic diagram in Figure 1 could use a lot more details, especially regarding the source of uncertainties (at least those have been considered in the model)

2. Figure2D and E. This is related to major comment 2. Please explain how the “ground truth” is chosen/defined and used in the simulation. Also, why “68%” is used ? The number just seems weird.

3.Figure6F. It would be more prefessional to connect the two lines reaching y=1.0 early on, to the top right corner of the plot area (ECD = 1 across that range).

4.Line 310, “This loses the subtle differences due to biotic and abiotic batch effects, effectively halving the data that goes into our inference problem” I don’t understand this, please elaborate.

5. I am unable to see the name of the particular section whenever the author refers to one (for example, line 53, "... See Section ?? for details...", also in line 56, etc.). I have tried two PCs (both using Acrobat on Win11). Are there invisible symbols due to PDF transformation?

Reviewer #3: In Razo Mejia et al, the authors describe a Bayesian framework for the analysis of barcode fitness assays. The authors show how general the framework can be, and test the validity of the inference on synthetic data. I really enjoyed the reading and seeing this type of inference approach finally seeing the light is a blessing to the community. I particularly enjoyed the section dealing with hyperparameters and how it can be used for interesting assays. While it is not the first time barcode fitness inferences have taken a Bayesian spin, it is the first to do so with a true Bayesian perspective on many different parameters. Unfortunately, I am not familiar with Julia (and don’t have it installed) and so cannot evaluate the software within the allotted review timeframe as the documentations are fairly lengthy.

The only major issue I have is that I am unable to judge how much this framework improves on the inference compared to the general approaches the community have undertaken. How poor are the current approaches implemented by the community? Or is it less so about the fitness estimates in the simple case (single environment, assuming replicates are same fitnesses), and more about the fact that the approach can be expanded? Or is it just the credible intervals on fitness values (augmented by priors)? Because my understanding is that almost all the ad hoc approaches taken currently by the community (by the lab of Levy, Petrov, Sherlock, Desai, Dunham, Gresham, or even earlier by the genomic era through the barcoded yeast deletion collection). are very adequate for their purposes. It would have been interesting and borderline necessary to compare at least the simple approaches to this. If it is simply differences between confidence intervals and credible intervals, then it would be ideal if the authors show distinct differences in their output notwithstanding the philosophical differences between them as the differences are usually fairly minimal when priors are not strong.

I have a few minor comments:

1) There are a few missing references to specific sections (e.g. line 53, line 56, but there are at least dozens throughout).

2) I’m very familiar with the experimental setup, but I think the readers will not appreciate why the unlabeled reference strain should be at ~90% of the population in these assays as described in lines 69-75. Many barcode assays have not done this in the literature and as far as I know there have been no reported issues. From my recollection, this is done due to frequency-dependent selection on some key lineages, but there is no evidence that this design resolves this (and frequency-dependent selection is not solved by any inference approach).

3) Is the simulation really sufficient for this work? Without going overboard and testing every single scenarios, it seems overly simple for the power of the proposed approach. I guess I see drift being implemented through the Poisson noise but is the Gaussian noise really adequate for the read frequency if one does not sequence to extremely high coverage? I’m also wondering if systematic noise that all play a role influence the framework at all: such as prolonged lag phase leading to uncertainty on the generations per transfer, exponential jackpotting issues during sequencing that may be poorly estimated from the neutral lineages, day to day variations etc. The main issue is that I feel like all the older approaches can adequately infer fitness for the simulation that was performed.

4) I think the aside debating the fundamental differences between confidence intervals and credible intervals is a bit misplaced. Yet if the authors want to keep this in, then perhaps the frequentist confidence intervals should be made more precise by emphasizing the repeated construction of CIs: “frequentist CIs represent the rangeS (emphasis on plural) of values where X% of ranges …” since upon repeating the experiments the confidence intervals would differ. As is written (in singular ‘range’), it may imply a fixed range of values between repetitions (vs a fixed construction method) and the ‘repetition’ containing the true population parameter would not really make sense.

**Have the authors made all data and (if applicable) computational code underlying the findings in their manuscript fully available?**

Reviewer #1: Yes

Reviewer #2: Yes

Reviewer #3: Yes

PLOS authors have the option to publish the peer review history of their article (what does this mean?). If published, this will include your full peer review and any attached files.

Reviewer #1: No

Reviewer #2: No

Reviewer #3: No
---

## [Decision Letter · Decision Letter 1]

21 Feb 2024

Dear Dr. Razo-Mejia,

We are pleased to inform you that your manuscript 'Bayesian inference of relative fitness on high-throughput pooled competition assays' has been provisionally accepted for publication in PLOS Computational Biology.

Best regards,

Sergei Maslov

Academic Editor

PLOS Computational Biology

Zhaolei Zhang

Section Editor

PLOS Computational Biology

Reviewer's Responses to Questions

**Comments to the Authors:**

Reviewer #1: Thanks to the authors for a very thorough and thoughtful response and revision. The analysis of the Kinsler dataset greatly adds to the paper, and I also appreciate their clarification of the flexibility of the model to handle non-logistic growth. I have no further comments.

Reviewer #2: In general, I am satisfied with the author's response, especially after they applied their method to empirical datasets. My congratulations go out to the authors for their nice work.

**Have the authors made all data and (if applicable) computational code underlying the findings in their manuscript fully available?**

Reviewer #1: Yes

Reviewer #2: Yes

PLOS authors have the option to publish the peer review history of their article (what does this mean?). If published, this will include your full peer review and any attached files.

Reviewer #1: No

Reviewer #2: No

---

## [Editor Report · Acceptance letter]

11 Mar 2024

PCOMPBIOL-D-23-01682R1 

Bayesian inference of relative fitness on high-throughput pooled competition assays

Dear Dr Razo-Mejia,

I am pleased to inform you that your manuscript has been formally accepted for publication in PLOS Computational Biology. Your manuscript is now with our production department and you will be notified of the publication date in due course.

With kind regards,

Lilla Horvath
